# Selenium Prevents Inflammation in Human Placenta and Adipose Tissue In Vitro: Implications for Metabolic Diseases of Pregnancy Associated with Inflammation

**DOI:** 10.3390/nu14163286

**Published:** 2022-08-11

**Authors:** Caitlyn Nguyen-Ngo, Anthony V. Perkins, Martha Lappas

**Affiliations:** 1Obstetrics, Nutrition and Endocrinology Group, Department of Obstetrics and Gynaecology, University of Melbourne, Parkville 3010, Australia; 2Mercy Perinatal Research Centre, Melbourne 3084, Australia; 3School of Pharmacy and Medical Sciences, Gold Coast Campus, Griffith University, Adelaide 9726, Australia

**Keywords:** selenium, placenta, adipose tissue, inflammation, insulin resistance, gestational diabetes, maternal obesity

## Abstract

Gestational diabetes mellitus (GDM) and maternal obesity are significant metabolic complications increasingly prevalent in pregnancy. Of major concern, both GDM and maternal obesity can have long-term detrimental impacts on the health of both mother and offspring. Recent research has shown that increased inflammation and oxidative stress are two features central to the pathophysiology of these metabolic conditions. Evidence suggests selenium supplementation may be linked to disease prevention in pregnancy; however, the specific effects of selenium on inflammation and oxidative stress associated with GDM and maternal obesity are unknown. Therefore, this study aimed to investigate the effect of selenium supplementation on an in vitro model of GDM and maternal obesity. Human placental tissue, visceral adipose tissue (VAT) and subcutaneous adipose tissue (SAT) were stimulated with either the bacterial product lipopolysaccharide (LPS) or the pro-inflammatory cytokine TNF-α. Selenium pre-treatment blocked LPS and TNF-α induced mRNA expression and secretion of pro-inflammatory cytokines and chemokines, while increasing anti-inflammatory cytokine and antioxidant mRNA expression in placenta, VAT and SAT. Selenium pre-treatment was also found to inhibit LPS- and TNF-α induced phosphorylation of ERK in placenta, VAT and SAT. These findings indicate that selenium may be able to prevent inflammation and oxidative stress associated with GDM and maternal obesity. Additional in vivo studies are required to identify the efficacy of selenium supplementation in preventing inflammatory pathways activated by GDM and maternal obesity and to elucidate the mechanism involved.

## 1. Introduction

The world is witnessing an alarming increase in gestational diabetes mellitus (GDM) which correlates with the current obesity epidemic. GDM [1] and maternal obesity in pregnancy [2] complicate between 15–20% of all pregnancies globally, compromising the health of women and ultimately that of the next generation. In Australia, 15–20% of all pregnancies are complicated by GDM [3], while the incidence of GDM among obese pregnant women is up 35% [4,5]. The long-term health risks for mothers include increased rates of type 2 diabetes [6] and cardiovascular disease (CVD), whereas their offspring have an unacceptably high lifelong risk of obesity, diabetes, CVD, and certain cancers later in life [7,8]. This represents a significant burden on healthcare resources [9,10], so a safe and effective intervention for GDM and maternal obesity that can reduce the long-term disease burden is critically needed.

GDM and maternal obesity are characterised by enhanced inflammation [11,12,13,14] and evidence of increased oxidative stress [15]. We have previously shown that placenta and adipose tissue respond to inflammatory insults, such as the pro-inflammatory cytokine TNF-α and the bacterial product lipopolysaccharide (LPS), by enhancing the production and expression of pro-inflammatory cytokines and chemokines [16]. Notably, pro-inflammatory mediators can induce maternal hyperglycaemia during GDM by reducing whole-body insulin sensitivity and glucose utilisation [17,18,19], by disrupting insulin signalling pathways in skeletal muscle and white adipose tissue [18,19,20,21,22,23,24]. In addition, pro-inflammatory mediators contribute to other key features of pregnancies with GDM and maternal obesity, such as increased oxidative and endoplasmic reticulum stress [15,25] and alterations in placental nutrient transport [26,27]. Given the crucial role of inflammation in GDM and maternal obesity, the potential for agents that possess anti-inflammatory and antioxidant properties as therapeutics for these metabolic diseases of pregnancy should be examined.

Selenium is an essential trace element that is important for human health. It possesses various biological functions, including acting as a potent anti-inflammatory and antioxidant agent [28]. Selenium exists in nature in organic forms (selenomethionine) and inorganic forms (selenite and selenate). It is naturally present in many foods (nuts and seeds, whole grains and dairy products, fish and meat), is added to others (e.g., cereals and bread), and is available as a dietary supplement often in organic and inorganic forms. Selenium is important for maternal health and foetal development during pregnancy. A series of studies have shown that selenium concentrations decrease during pregnancy, and most dietary advice includes an increased requirement for selenium during gestation [29]. There are now several lines of evidence that link selenium supplementation to disease prevention in pregnancy [30]. Notably, epidemiological and clinical studies have shown that selenium deficiency in early pregnancy is a risk factor for GDM [31,32]. Furthermore, selenium supplementation in pregnant women with GDM improves glucose homeostasis [33]. Recent studies have also shown that maternal selenium deficiency during pregnancy in mice adversely alters placental function [34]. Together the available data strongly suggests that selenium may be an exciting and novel therapeutic for GDM and maternal obesity and improving maternal and offspring health outcomes.

Selenium has been shown to possess anti-inflammatory and antioxidant properties in various inflammatory-based in vitro and in vivo disease models [28,35,36,37,38,39,40,41,42,43,44,45,46,47]. There are, however, no studies that have assessed the anti-inflammatory and antioxidant effects of selenium in human placenta and adipose tissue. We hypothesise that selenium supplementation will reduce pro-inflammatory cytokines and chemokines and increase anti-inflammatory cytokines and antioxidants in placenta and adipose tissue induced by TNF-α or LPS.

## 2. Materials and Methods

### 2.1. Ethical Approvals

The Mercy Hospital for Women Research and Ethics Committee (Mercy Health, Ethics approval number R04-29) approved this study. Written informed consent was obtained from all participating women.

### 2.2. Tissue Collection

Placental tissue, subcutaneous adipose tissue (SAT) and visceral adipose tissue (VAT) were obtained from healthy normal glucose tolerant (NGT) women with a BMI < 30 kg/m^2^ who were delivering healthy, singleton infants at term (37–41 weeks of gestation) via elective Caesarean section in the absence of labour. The exclusion criteria were as previously described [48] and included women with any vascular/renal complications, multiple gestations, GDM, asthma, smokers, preeclampsia, chorioamnionitis, placental abruption, acute foetal distress, and women with any other adverse underlying medical conditions. All tissues were processed within 15 min of the Caesarean delivery.

### 2.3. Tissue Explants

Tissue explants, prepared as previously described [49,50], were used to determine the effect of selenium on the mRNA expression and protein release of pro-inflammatory cytokines and chemokines in placenta, SAT and VAT. In this study, sodium selenite was used as a source of selenium. Briefly, placenta, VAT and SAT were washed in ice-cold PBS and blunt dissected to remove visible connective tissue, vessels and calcium deposits. The processed tissues were then pre-incubated for 1 hr in Dulbecco’s Modified Eagle Media (DMEM) (containing 100 U/mL penicillin G and 100 µg/mL streptomycin) at 37 °C in a humidified incubator of 5% CO_2_ and 21% O_2_ (VAT and SAT) or 8% O_2_ (placenta). Tissues were then pre-incubated with 10 μM selenium (sodium selenite; ≥98% pure; Sigma-Aldrich; St. Louis, MO, USA) for 1 h, and then treated with 10 µg/mL LPS (derived from *Escherichia coli* 026:B6; Sigma-Aldrich) or 10 ng/mL TNF-α (PeproTech) for a further 20 h. The final concentration of selenium chosen was based on previously published studies [38,51,52,53] and an initial dose-response study. After 20 h incubation with selenium, tissues and conditioned media were collected separately and stored at −80 °C for analysis by RT-qPCR or ELISA, as detailed below.

We also tested the mechanism through which selenium may act by examining its effect on the MAPK signalling pathway protein ERK. For these studies, tissues were incubated with 10 μM selenium for 20 h and then treated with 10 ng/mL TNF-α or 10 µg/mL LPS for 15 min. Tissues were then collected and stored at −80 °C for analysis by Western blotting, as detailed below.

### 2.4. Enzyme Immunoassays

The levels of GM-CSF, IL1A, IL1B, IL6, CCL2, CCL4, CXCL1, CXCL5 and CXCL8 in the conditioned media were measured by sandwich ELISA from R&D Systems (Minneapolis, MN, USA) per the manufacturer’s instructions. The interassay and intraassay coefficients of variation for all assays were consistently less than 10%.

### 2.5. Quantitative RT-PCR (RT-qPCR)

RNA extractions, cDNA synthesis and RT-qPCR were performed as previously described [51] using 100 nM of pre-designed and validated QuantiTect primers (primer sequences not available) (Qiagen, Clayton, VIC, Australia). The primers catalogue number are: GM-CSF, QT00000896; IL1A, QT00001127; IL1B, QT00021385; IL4, QT00012565; IL6, QT00083720; IL13, QT00000511; CCL2, QT00212730; CCL3, QT01008063; CCL4, QT01008070; CCL8, QT00212639; CXCL1, QT00199752; CXCL2, QT00013104; CXCL5, QT00203686; CXCL8, QT00000322; CXCL10, QT01003065; GPx, QT00203392; SDHA, QT00059486; TrxR, QT00055902; and YWHAZ, QT00087962. Target gene Ct values were normalised to the average YWHAZ and SDHA Ct values of the same cDNA sample. Fold differences were determined using the comparative Ct method.

### 2.6. Western Blotting

Western blotting was used to determine the effect of selenium on phosphorylation (i.e., activation) of the MAPK signalling pathway protein ERK. Western blotting was performed as previously described [52]. Mouse monoclonal pERK (sc-7383; Santa Cruz Biotechnology) and rabbit polyclonal ERK (sc-93; Santa Cruz Biotechnology, Dalla, Texas, USA) were used at 0.2 μg/mL. Membranes were viewed and analysed using the ChemiDoc MP system (Bio-Rad Laboratories; Gladesville, NSW, Australia). Semi-quantitative analysis of the relative density of the bands in Western blots was performed using Quantity One 4.2.1 image analysis software (Bio-Rad Laboratories, Hercules, CA, USA).

### 2.7. Statistical Analysis

Tissue explants were analysed from six patients. All statistical analyses were undertaken using GraphPad Prism (GraphPad Software, La Jolla, CA, USA). Normality of the data was assessed using the Shapiro–Wilk test. Non-normalised data were logarithmically transformed before analysis by a repeated measures one-way ANOVA (with LSD post hoc testing to discriminate among the means). Statistical significance was ascribed to a *p*-value ≤ 0.05.

## 3. Results

### 3.1. Effect of Selenium on Pro-Inflammatory Cytokines Expression

The effect of selenium on LPS or TNF-α induced expression and secretion of pro-inflammatory cytokines in placenta and adipose tissue (VAT and SAT) is illustrated in Figure 1 and Figure 2, respectively. The levels of GM-CSF and IL1-B in the incubation media of placenta, VAT and SAT, and the levels of IL1-A in the conditioned media from VAT and SAT tissues was below the sensitivity of the assay and therefore not assessed.

In placental tissue, LPS and TNF-α treatment significantly increased GM-CSF, IL1-A, IL1-B and IL6 mRNA expression (Figure 1A,B,D,E), and IL1-A and IL6 secretion (Figure 1C,F). Selenium pre-treatment significantly suppressed LPS and TNF-α induced IL1-A, IL1-B and IL6 mRNA expression and IL1-A and IL6 secretion (Figure 1B–F). Selenium pre-treatment also significantly reduced TNF-α induced GM-CSF mRNA expression (Figure 1A); however, there was no effect of selenium on LPS-induced GM-CSF mRNA expression (Figure 1A).

In VAT, LPS and TNF-α induced significant increases in GM-CSF, IL1-A, IL1-B and IL6 mRNA expression (Figure 2A–D) and IL6 secretion (Figure 2E). Selenium pre-treatment significantly suppressed LPS and TNF-induced IL1-A, IL1-B and IL6 mRNA expression and IL6 protein release (Figure 2B–E). Selenium pre-treatment also significantly reduced TNF-induced GM-CSF mRNA expression (Figure 2A); however, there was no effect of selenium on LPS-induced GM-CSF mRNA expression (Figure 2A).

In SAT, LPS and TNF-α significantly upregulated GM-CSF, IL1-A, IL1-B and IL6 mRNA expression (Figure 2F–I) and IL6 secretion (Figure 2J). Selenium pre-treatment significantly down-regulated LPS- and TNF-α induced IL1-A, IL1-B and IL6 mRNA expression (Figure 2G–I). Selenium pre-treatment also significantly reduced GM-CSF mRNA expression and IL6 secretion in the presence of LPS (Figure 2F,J) but not TNF-α (Figure 2F,J).

### 3.2. Effect of Selenium on Anti-Inflammatory Cytokine Expression

The effect of selenium on LPS or TNF-α induced expression of anti-inflammatory cytokines in placenta and adipose tissue (VAT and SAT) is presented in Figure 3. There was no effect of LPS or TNF-α treatment on IL4 or IL13 mRNA expression in placenta (Figure 3A,B), VAT (Figure 3C,D) or SAT (Figure 3E,F). In placenta, selenium pre-treatment significantly increased IL4 and IL13 mRNA expression in the presence of LPS; there was, however, no effect of selenium on IL4 and IL13 mRNA expression in the presence of TNF-α (Figure 3A,B). In VAT, selenium pre-treatment significantly increased IL4 and IL13 mRNA expression in the presence of LPS and TNF-α (Figure 3C,D). In SAT, selenium pre-treatment significantly upregulated IL13 mRNA expression in the presence of LPS or TNF-α (Figure 3F). There was, however, no effect of selenium pre-treatment on IL4 mRNA expression in the presence of LPS or TNF-α (Figure 3E).

### 3.3. Effect of Selenium on Chemokine Expression

The effect of selenium pre-treatment on LPS- or TNF-α induced expression and secretion of CCL and CXCL chemokines in placenta is demonstrated in Figure 4 and Figure 5, in VAT in Figure 6 and Figure 7, and in SAT in Figure 8 and Figure 9. In all three tissues, the levels of CCL3, CCL8, CXCL2 and CXCL10 in the conditioned media were below the sensitivity of the assay and therefore not assessed.

In placenta, LPS and TNF-α significantly increased CCL2, CCL3, CCL4, and CCL8 mRNA expression (Figure 4A,C,D,F) and CCL2 and CCL4 secretion (Figure 4B,E). Likewise, LPS and TNF-α treatment significantly upregulated CXCL1, CXCL5 and CXCL8 mRNA expression and secretion (Figure 5A,B,D–G). LPS treatment also significantly upregulated CXCL2 and CXCL10 mRNA expression; however, there was no effect of TNF-α treatment on CXCL2 or CXCL10 mRNA expression (Figure 5C,H). Selenium pre-treatment significantly reduced LPS- and TNF-α induced CCL3, CCL4 mRNA expression (Figure 4C,D) and CCL2 and CCL4 secretion (Figure 4B,E); and CXCL1, CXCL5 and CXCL8 mRNA expression and secretion (Figure 5A,B,D–G). Selenium pre-treatment also significantly reduced TNF-α-induced CCL2 mRNA expression (Figure 4A). There was no effect of selenium pre-treatment on LPS-induced CCL2 mRNA expression (Figure 4A), or LPS- and TNF-α induced CCL8, CXCL2 and CXCL10 mRNA expression (Figure 4F and Figure 5C,H).

In VAT, LPS significantly increased CCL2, CCL3, CCL4, CCL8, CXCL1, CXCL2, CXCL5, CXCL8 and CXCL10 mRNA expression, and CCL2, CCL4, CXCL1 and CXCL8 secretion (Figure 6 and Figure 7). TNF-α treatment augmented CCL2, CCL4, CCL8, CXCL1, CXCL5, CXCL8 and CXCL10 mRNA expression, and CCL2, CCL4, CXCL1, CXCL5 and CXCL8 secretion (Figure 6 and Figure 7). There was no effect of LPS on CXCL5 concentration (Figure 7E) or TNF-α on CCL3 and CXCL2 mRNA expression (Figure 6C and Figure 7C). Selenium pre-treatment significantly suppressed LPS-induced CCL2, CCL8, CXCL1, CXCL2, CXCL5, CXCL8 and CXCL10 mRNA expression (Figure 6A,F and Figure 7A,C,D,F,H), and CCL2, CXCL1, CXCL5 and CXCL8 secretion (Figure 6B and Figure 7B,E,G). Selenium pre-treatment also significantly reduced TNF-α-induced CCL2 and CCL4 mRNA expression and secretion (Figure 6A,B,D,E). There was, however, no effect of selenium pre-treatment on LPS-induced CCL3 and CCL4 mRNA expression and secretion (Figure 6C–E) or TNF-α-induced CCL3 and CCL8 mRNA expression (Figure 6C,F). Selenium also significantly reduced LPS-induced CXCL8 mRNA expression; however, there was no effect on TNF-α-induced CXCL8 mRNA expression (Figure 7F).

In SAT, LPS and TNF-α significantly increased CCL2, CCL3, CCL4 and CCL8 mRNA expression (Figure 8A,C,D,F) and CCL2 secretion (Figure 8B). LPS significantly increased CCL4 secretion; however, there was no effect of TNF-α on CCL4 secretion (Figure 8E). LPS and TNF-α also significantly increased the mRNA expression and secretion of all CXCL chemokines (Figure 9). Selenium pre-treatment downregulated LPS-and TNF-α induced CCL2, CCL3, CCL4 and CCL8 mRNA expression (Figure 8A,C,D,F) and CCL2 secretion (Figure 8B). Selenium also downregulated TNF-α-induced CCL4 secretion; however, there was no effect of selenium pre-treatment on LPS-induced CCL4 secretion (Figure 8E). Selenium treatment also downregulated the mRNA expression and secretion of all LPS- and TNF-α induced CXCL chemokines (Figure 9).

### 3.4. Effect of Selenium on Selenoprotein Expression

The effect of selenium on the mRNA expression on antioxidant selenoenzymes glutathione peroxidase (GPx) and thioredoxin reductase (TrxR) in placenta, VAT and SAT is illustrated in Figure 10. In placenta, there was no effect of LPS or TNF-α treatment on GPx or TrxR mRNA expression (Figure 10A,B). Selenium pre-treatment significantly upregulated GPx mRNA expression in the presence of LPS; however, there was no effect of selenium on GPx mRNA expression in the presence of TNF-α (Figure 10A). There was also no effect of selenium on TrxR mRNA expression in the presence of LPS or TNF-α (Figure 10B).

In VAT, LPS treatment significantly downregulated GPx and TrxR mRNA expression (Figure 10C,D). TNF-α treatment also significantly downregulated TrxR mRNA expression (Figure 10D), but there was no effect of TNF-α treatment on GPx mRNA expression (Figure 10C). Selenium pre-treatment significantly increased LPS-impaired GPx mRNA expression (Figure 10D) and LPS- and TNF-α impaired TrxR mRNA expression (Figure 10D). In SAT, there was no effect of LPS or TNF treatment on GPx or TrxR mRNA expression (Figure 10E,F). Selenium pre-treatment, however, significantly increased GPx and TrxR mRNA expression in the presence of LPS or TNF-α (Figure 10E,F).

### 3.5. Selenium Pre-Treatment Inhibits Activation of the MAPK Protein ERK

Previously, we demonstrated that selenium treatment inhibited ERK activation in foetal membranes and myometrium stimulated with LPS [38]. Therefore, in this study we sought to identify whether selenium may also target ERK activation in placenta, VAT and SAT stimulated with LPS or TNF-α. In placenta and VAT, treatment with LPS or TNF-α significantly upregulated pERK protein expression (Figure 11A,B). TNF-α treatment also significantly increased pERK protein expression in SAT; however, there was no effect of LPS treatment on pERK protein expression (Figure 11C). In placenta, selenium pre-treatment significantly downregulated pERK protein expression in the presence of TNF-α but not LPS (Figure 11A). In VAT and SAT, selenium pre-treatment significantly suppressed LPS and TNF-α-induced pERK protein expression (Figure 11B,C).

## 4. Discussion

For the first time, the data presented in this study demonstrates the effects of selenium pre-treatment on maternal inflammation and oxidative stress associated with GDM and maternal obesity. To induce an inflammatory environment akin to GDM and maternal obesity, LPS and TNF-α were used to treat placenta and adipose tissue (visceral and subcutaneous) in the absence or presence of selenium. Here, we show that selenium pre-treatment significantly decreased inflammation-induced expression of pro-inflammatory mediators in human placenta and adipose tissue obtained from pregnant women. Selenium pre-treatment also upregulated mRNA expression of anti-inflammatory cytokines and antioxidant selenoenzymes in placenta and adipose tissue obtained from pregnant women. Finally, selenium pre-treatment blocked inflammation-induced activation of MAPK ERK signalling in human adipose tissue obtained from pregnant women.

GDM and maternal obesity are characterised by low-grade maternal inflammation, which is thought to underlie other key features of GDM and maternal obesity [16,54,55,56]. Non-infectious inflammation or infectious insults induce a pro-inflammatory response by stimulating production of pro-inflammatory cytokines and chemokines in placenta [17,57,58,59,60] and maternal adipose tissue [61,62,63]. Chemokines exist as two main subfamilies, CCL and CXCL, and activate maternal peripheral leukocytes to induce their infiltration into tissues. This leukocytic infiltrate is a key source of pro-inflammatory cytokines that can contribute to the development of GDM and maternal obesity. This includes inflammation-induced defects in insulin signalling and glucose uptake in adipose tissue [20,64] and skeletal muscle [18,65], alterations in placental nutrient transport [26,27], and increased oxidative and endoplasmic reticulum stress [15,25]. In this study, placenta and adipose tissue (visceral and subcutaneous) from normal pregnant women were stimulated with the bacterial product LPS or the pro-inflammatory cytokine TNF-α to induce an inflammatory state akin to GDM or maternal obesity. Selenium pre-treatment significantly suppressed LPS and TNF-α induced pro-inflammatory cytokine and chemokine mRNA expression and protein secretion in placenta, VAT and SAT. Selenium pre-treatment also promoted mRNA expression of anti-inflammatory cytokines in placenta, VAT and SAT. Of note, there were also tissue-specific effects of selenium. For example, selenium decreased CXCL2 and CXCL10 mRNA expression in VAT and SAT, but not in placenta. Higher doses of selenium and a different incubation time may be needed to observe similar effects. Additionally, for most endpoints, the mRNA and protein secretion were similar; however, there were a few exceptions. For example, in SAT, selenium decreased TNF-α induced IL6 mRNA expression but had no effect on TNF-α induced IL6 secretion. Post-translational modifications, differential regulation of transcription and translation may account for these differences. Regardless, taken together, our findings indicate that selenium may block production of pro-inflammatory cytokines and chemokines in favour of promoting an anti-inflammatory response in human placenta and adipose tissue.

The biochemical and cellular actions of selenium are mediated through its effects on selenoproteins with antioxidant properties such as the GPx and TrxR family [66]. In this study, selenium pre-treatment significantly upregulated mRNA expression of GPx and TrxR in placenta, VAT and SAT. Notably, however, tissue-specific actions were observed with selenium exerting more potent antioxidant actions in VAT and SAT compared to placenta. Selenite, which is an inorganic form of selenium, was used in this study. It is important to note that selenium can also exist in nature in organic forms such as selenomethionine and selenocysteine and they have differing effects on selenoproteins. Nevertheless, these findings are of significance as GDM and maternal obesity are characterised by oxidative stress [15,67]. This includes higher concentrations of the oxidative stress biomarker 8-isoprostane [58] and reduced antioxidant activity [68,69] in placenta, adipose tissue and skeletal muscle. These findings suggest that selenium may be able to control oxidative stress by promoting antioxidant expression. Further studies assessing the effect of selenium on oxidative stress markers may confirm the antioxidant-promoting properties of this essential trace element.

The MAPK ERK signalling pathway is thought to contribute to the pathophysiology of both GDM and maternal obesity. Previous research demonstrates increased pERK expression in placentas from women with GDM compared to healthy pregnant women [70]. Notably, in vitro studies have shown that ERK inhibition prevented the inflammation-induced expression of pro-inflammatory cytokines and chemokines in placenta [71] and adipose tissue [72]. Selenium deficiency in mice with LPS-induced endometriosis has been shown to enhance ERK phosphorylation [43], while selenium supplementation inhibited ERK activation in foetal membranes and myometrium [38]. Likewise, in this study, selenium pre-treatment significantly blunted inflammation-induced ERK phosphorylation in placenta, VAT and SAT. Interestingly, in placenta, while selenium blocked TNF-α induced ERK phosphorylation, there was no effect of selenium pre-treatment on LPS induced ERK phosphorylation. Selenium is also known to exert it actions by suppressing the NF-κB pathway [73]. Selenium may therefore suppress LPS inflammation in the placenta by targeting other signalling pathways. Nevertheless, taken together, our results suggest that the anti-inflammatory and antioxidant actions of selenium are elicited through suppression of ERK activation.

## 5. Conclusions

In summary, selenium supplementation prevented inflammation and promoted antioxidant expression in placenta and maternal adipose tissue in an in vitro model of GDM and maternal obesity. Maternal and placental inflammation [14,17,55] and oxidative stress [15,58,68,69] are key features common to both GDM and maternal obesity. These findings are of further relevance given the role of inflammation in driving other key characteristics of GDM and maternal obesity, such as maternal peripheral insulin resistance [18,65] and alterations in placental nutrient transport [26,27]. Our novel and exciting results are in agreement with clinical studies demonstrating that low maternal selenium levels are associated with an increased risk of developing GDM [31,32] and selenium supplementation in pregnant women with GDM improves glucose homeostasis [33]. Recent studies in mice have also shown that maternal selenium deficiency during pregnancy adversely alters placental function [34]. Together these studies suggest that selenium supplementation may be an exciting and novel therapeutic to improve adverse maternal and offspring health outcomes associated with GDM and obesity.

## Figures and Tables

**Figure 1 nutrients-14-03286-f001:**
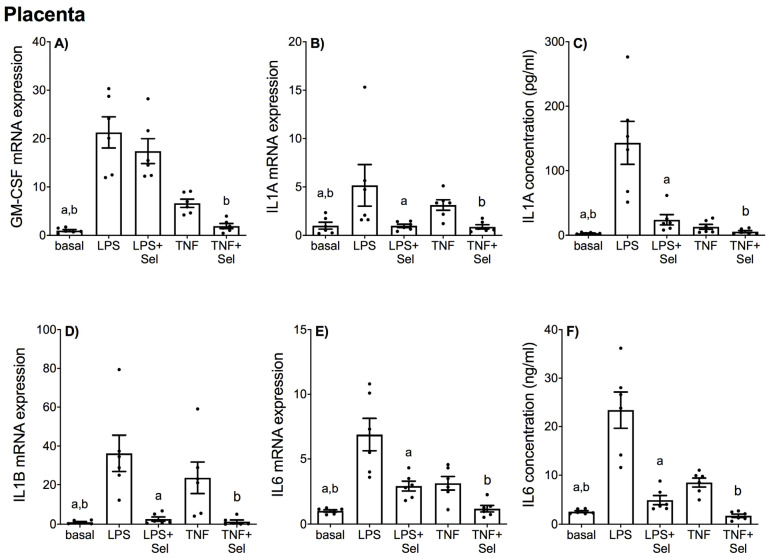
Effect of selenium on pro-inflammatory cytokine expression in human placenta. Human placenta was incubated with or without 10 μM sodium selenite (Sel) for 1 h and then stimulated with 10 µg/mL LPS or 10 ng/mL TNF-α for a further 20 h (*n* = 6 patients). (**A**,**B**,**D**,**E**) GM-CSF, IL1-A, IL1-B and IL6 mRNA expression were analysed by RT-qPCR and fold change was calculated relative to basal expression. (**C**,**F**) The concentration of IL1-A and IL6 in the conditioned media was assayed by ELISA. For all graphs, individual data points represent six independent experiments and are displayed as mean ± SEM. ^a^
*p* ≤ 0.05 vs. LPS, ^b^
*p* ≤ 0.05 vs. TNF-α; repeated measures one-way ANOVA.

**Figure 2 nutrients-14-03286-f002:**
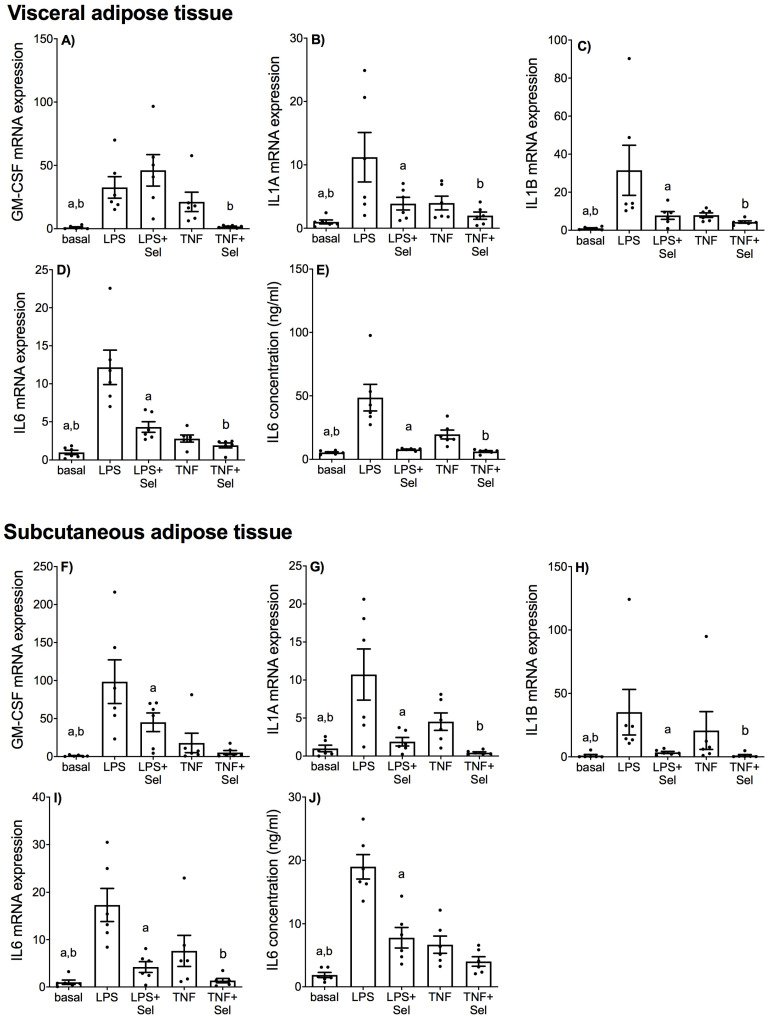
Effect of selenium on pro-inflammatory cytokine expression in VAT and SAT from pregnant women. (**A**–**E**) VAT (*n* = 6 patients) and (**F**–**J**) SAT (*n* = 6 patients) were incubated with or without 10 μM sodium selenite (Sel) for 1 h and then stimulated with 10 µg/mL LPS or 10 ng/mL TNF-α for a further 20 h. (**A**–**D**,**F**–**I**) GM-CSF, IL1-A, IL1-B and IL6 mRNA expression was analysed by RT-qPCR and fold change was calculated relative to basal expression. (**E**,**J**) The concentration of IL6 in the conditioned media was assayed by ELISA. For all graphs, individual data points represent six independent experiments and are displayed as mean ± SEM. ^a^
*p* ≤ 0.05 vs. LPS, ^b^
*p* ≤ 0.05 vs. TNF; repeated measures one-way ANOVA.

**Figure 3 nutrients-14-03286-f003:**
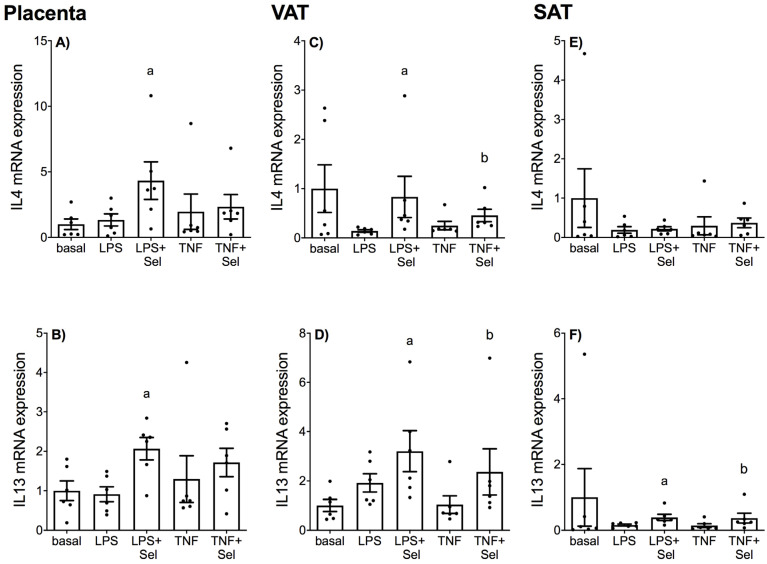
Effect of selenium on anti-inflammatory cytokine expression in human placenta, and VAT and SAT from pregnant women. (**A**,**B**) Placenta (*n* = 6 patients), (**C**,**D**) VAT (*n* = 6 patients) and (**E**,**F**) SAT (*n* = 6 patients) were incubated with or without 10 μM sodium selenite (Sel) for 1 h and then stimulated with 10 µg/mL LPS or 10 ng/mL TNF-α for a further 20 h. IL4 and IL13 mRNA expression was analysed by RT-qPCR and fold change was calculated relative to basal expression. For all graphs, individual data points represent 6 independent experiments and are displayed as mean ± SEM. ^a^
*p* ≤ 0.05 vs. LPS, ^b^
*p* ≤ 0.05 vs. TNF; repeated measures one-way ANOVA.

**Figure 4 nutrients-14-03286-f004:**
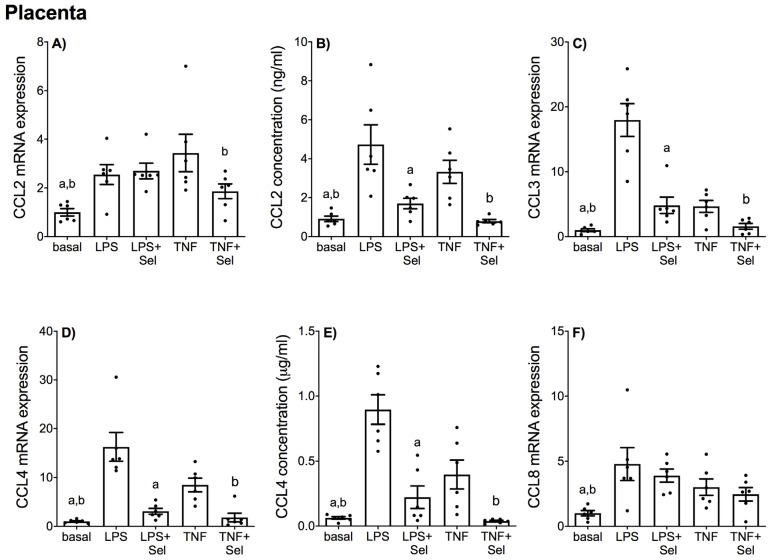
Effect of selenium on CCL chemokine expression in human placenta. Placenta was incubated with or without 10 μM sodium selenite (Sel) for 1 h and then stimulated with 10 µg/mL LPS or 10 ng/mL TNF-α for a further 20 h (*n* = 6 patients). (**A**,**C**,**D**,**F**) CCL2, CCL3, CCL4 and CCL8 mRNA expression was analysed by RT-qPCR and fold change was calculated relative to basal expression. (**B**,**E**) The concentration of CCL2 and CCL4 in the conditioned media was assayed by ELISA. For all graphs, individual data points represent six independent experiments and are displayed as mean ± SEM. ^a^
*p* ≤ 0.05 vs. LPS, ^b^
*p* ≤ 0.05 vs. TNF; repeated measures one-way ANOVA.

**Figure 5 nutrients-14-03286-f005:**
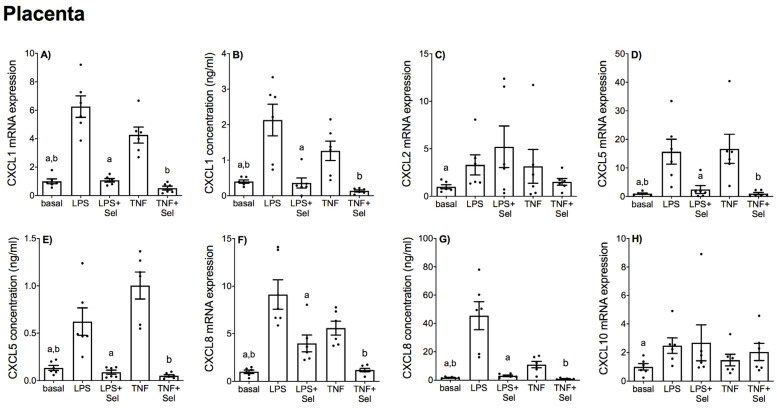
Effect of selenium on CXCL chemokine expression in human placenta. Placenta was incubated with or without 10 μM sodium selenite (Sel) for 1 h and then stimulated with 10 µg/mL LPS or 10 ng/mL TNF-α for a further 20 h (*n* = 6 patients). (**A**,**C**,**D**,**F**,**H**) CXCL1, CXCL2, CXCL5, CXCL8 and CXCL10 mRNA expression was analysed by RT-qPCR and fold change was calculated relative to basal expression. (**B**,**E**,**G**) The concentration of CXCL1, CXCL5 and CXCL8 in the conditioned media was assayed by ELISA. For all graphs, individual data points represent six independent experiments and are displayed as mean ± SEM. ^a^
*p* ≤ 0.05 vs. LPS, ^b^
*p* ≤ 0.05 vs. TNF; repeated measures one-way ANOVA.

**Figure 6 nutrients-14-03286-f006:**
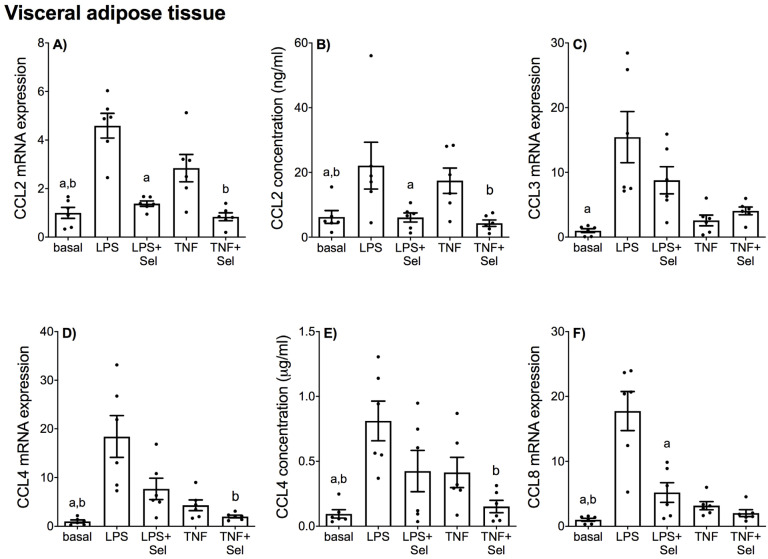
Effect of selenium on CCL chemokine expression in VAT from pregnant women. VAT was incubated with or without 10 μM sodium selenite (Sel) for 1 h and then stimulated with 10 µg/mL LPS or 10 ng/mL TNF-α for a further 20 h (*n* = 6 patients). (**A**,**C**,**D**,**F**) CCL2, CCL3, CCL4 and CCL8 mRNA expression was analysed by RT-qPCR and fold change was calculated relative to basal expression. (**B**,**E**) The concentration of CCL2 and CCL4 in the conditioned media was assayed by ELISA. For all graphs, individual data points represent six independent experiments and are displayed as mean ± SEM. ^a^
*p* ≤ 0.05 vs. LPS, ^b^
*p* ≤ 0.05 vs. TNF; repeated measures one-way ANOVA.

**Figure 7 nutrients-14-03286-f007:**
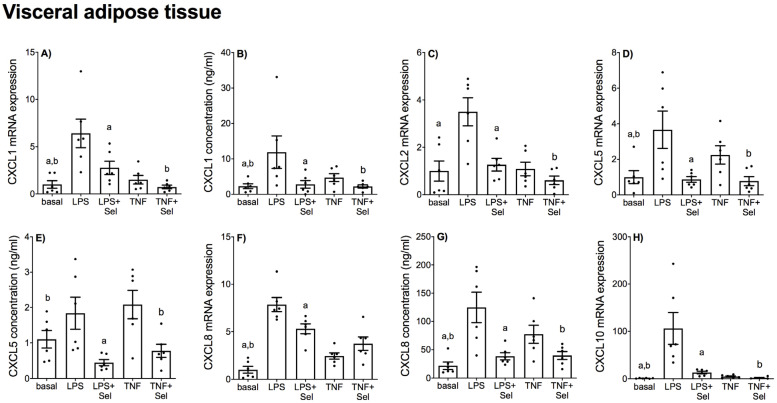
Effect of selenium on CXCL chemokine expression in VAT from pregnant women. VAT was incubated with or without 10 μM sodium selenite (Sel) for 1 h and then stimulated with 10 µg/mL LPS or 10 ng/mL TNF-α for a further 20 h (*n* = 6 patients). (**A**,**C**,**D**,**F**,**H**) CXCL1, CXCL2, CXCL5, CXCL8 and CXCL10 mRNA expression was analysed by RT-qPCR and fold change was calculated relative to basal expression. (**B**,**E**,**G**) The concentration of CXCL1, CXCL5 and CXCL8 in the conditioned media was assayed by ELISA. For all graphs, individual data points represent six independent experiments and are displayed as mean ± SEM. ^a^
*p* ≤ 0.05 vs. LPS, ^b^
*p* ≤ 0.05 vs. TNF; repeated measures one-way ANOVA.

**Figure 8 nutrients-14-03286-f008:**
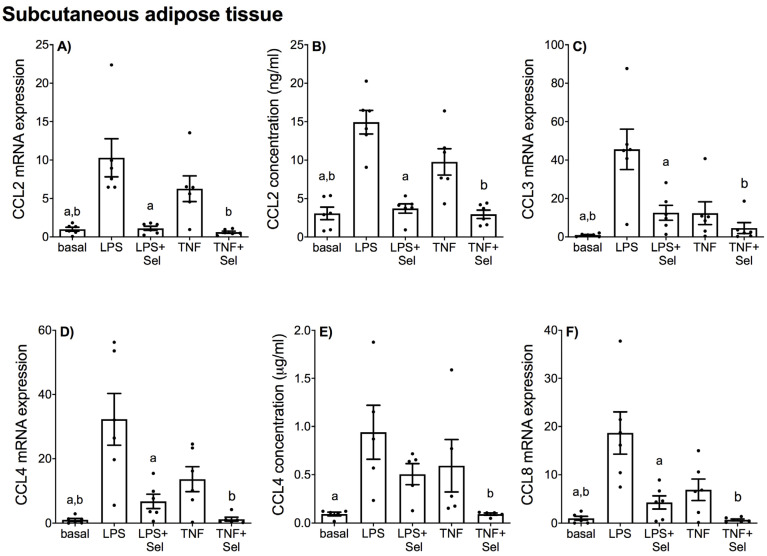
Effect of selenium on CCL chemokine expression in SAT from pregnant women. SAT was incubated with or without 10 μM sodium selenite (Sel) for 1 h and then stimulated with 10 µg/mL LPS or 10 ng/mL TNF-α for a further 20 h (*n* = 6 patients). (**A**,**C**,**D**,**F**) CCL2, CCL3, CCL4 and CCL8 mRNA expression was analysed by RT-qPCR and fold change was calculated relative to basal expression. (**B**,**E**) The concentration of CCL2 and CCL4 in the conditioned media was assayed by ELISA. For all graphs, individual data points represent six independent experiments and are displayed as mean ± SEM. ^a^
*p* ≤ 0.05 vs. LPS, ^b^
*p* ≤ 0.05 vs. TNF; repeated measures one-way ANOVA.

**Figure 9 nutrients-14-03286-f009:**
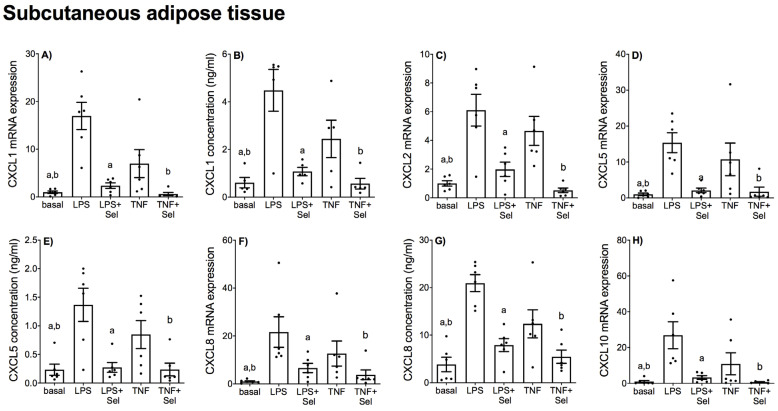
Effect of selenium on CXCL chemokine expression in SAT from pregnant women. SAT was incubated with or without 10 μM sodium selenite (Sel) for 1 h and then stimulated with 10 µg/mL LPS or 10 ng/mL TNF-α for a further 20 h (*n* = 6 patients). (**A**,**C**,**D**,**F**,**H**) CXCL1, CXCL2, CXCL5, CXCL8 and CXCL10 mRNA expression was analysed by RT-qPCR and fold change was calculated relative to basal expression. (**B**,**E**,**G**) The concentration of CXCL1, CXCL5 and CXCL8 in the conditioned media was assayed by ELISA. For all graphs, individual data points represent six independent experiments and are displayed as mean ± SEM. ^a^
*p* ≤ 0.05 vs. LPS, ^b^
*p* ≤ 0.05 vs. TNF; repeated measures one-way ANOVA.

**Figure 10 nutrients-14-03286-f010:**
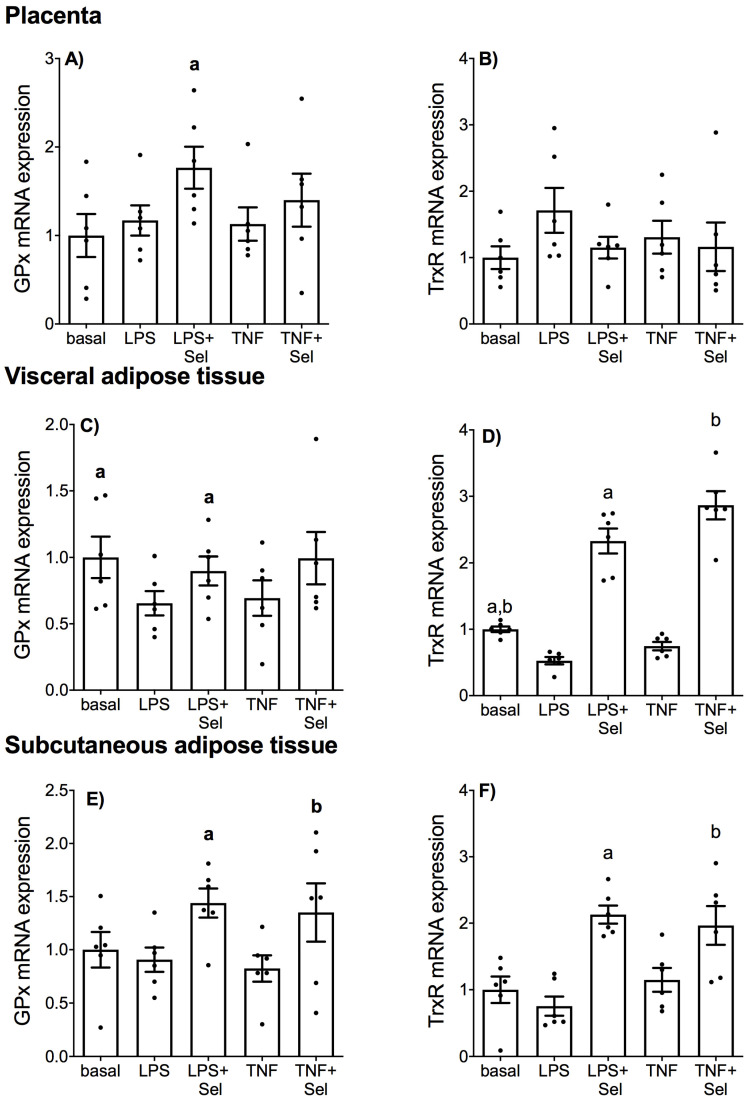
Effect of selenium on antioxidant selenoenzyme expression in human placenta, and VAT and SAT from pregnant women. (**A**,**B**) Placenta (*n* = 6 patients), (**C**,**D**) VAT (*n* = 6 patients) and (**E**,**F**) SAT (*n* = 6 patients) were incubated with or without 10 μM sodium selenite (Sel) for 1 h and then stimulated with 10 µg/mL LPS or 10 ng/mL TNF-α for a further 20 h. GPx and TrxR mRNA expression was analysed by RT-qPCR and fold change was calculated relative to basal expression. For all graphs, individual data points represent six independent experiments and are displayed as mean ± SEM. ^a^
*p* ≤ 0.05 vs. LPS, ^b^
*p* ≤ 0.05 vs. TNF; repeated measures one-way ANOVA.

**Figure 11 nutrients-14-03286-f011:**
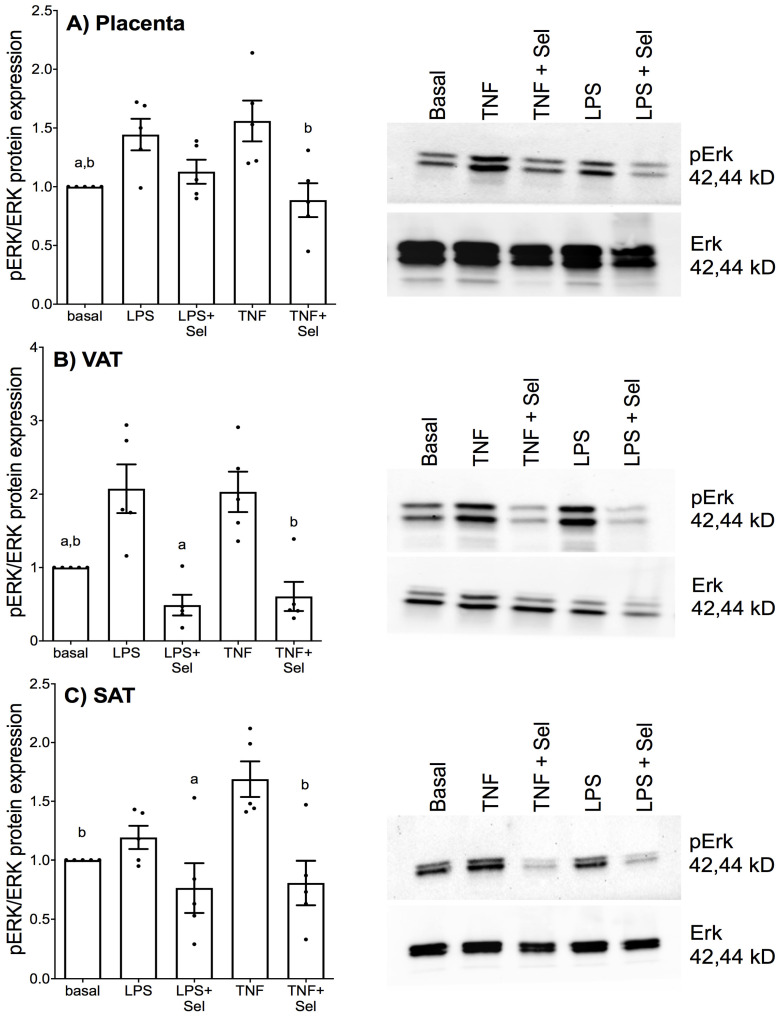
Effect of selenium on ERK activation in human placenta, and VAT and SAT from pregnant women. (**A**) Placenta (*n* = 5 patients), (**B**) VAT (*n* = 5 patients) and (**C**) SAT (*n* = 5 patients) were incubated with or without 10 μM sodium selenite (Sel) overnight and then stimulated with 10 µg/mL LPS or 10 ng/mL TNF-α for a further 15 min. The protein expression of phosphorylated ERK1 (pERK) and total ERK1 was analysed by Western blotting; pERK protein expression was normalised to total ERK expression. A Representative western blot from one patient is shown. For all graphs, the data were calculated relative to basal expression and displayed as mean ± SEM with data points representing five individual experiments. ^a^
*p* ≤ 0.05 vs. LPS, ^b^
*p* ≤ 0.05 vs. TNF; repeated measures one-way ANOVA.

## Data Availability

All data generated during this study will be made available as per the Data Availability Statements in the Section “MDPI Research Data Policies” at https://www.mdpi.com/ethics, accessed on 1 August 2022.

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
