# Peer review of "Selenium Prevents Inflammation in Human Placenta and Adipose Tissue In Vitro: Implications for Metabolic Diseases of Pregnancy Associated with Inflammation"

_nutrients, 2022, doi:10.3390/nu14163286_

Round 1

Reviewer 1 Report

Dear Authors,

Thank you for the opportunity to review the communication entitled „Selenium prevents inflammation in human placenta and adipose tissue: Implications for metabolic diseases of pregnancy 3 associated with inflammation” which  addresses the following objective:

-        selenium supplementation will reduce pro-inflammatory cytokines and chemokines and increase anti-inflammatory cytokines and antioxidants in placenta and adipose tissue induced by TNF-α or LPS.

The results of this study are important because they strengthens previous findings regarding the effects of selenium and also because of their novelty. Studies regarding the anti-inflamatory effect of nutrients are scarce and could represent a starting point in developing new therapies in diabetes, but also in other metabolic diseases.

The study is correctly designed and technically sound. The methods used in this research are well described and provide sufficient details to be understand. The research methodology is in line with the proposed objectives.

The results are appropriately interpreted and respond to the hypothesis of the study.

The disscutions address the findings of the research (line 352-362) and propose potential new explanation regarding the mediating role of selenoproteins with antioxidant properties such as the GPx and TrxR family.

I have some comments on the article, which are listed below:

1.     This study presents the results of selenium supplementation in vitro conditions – this aspect should be mentiones in the title of the article

2.     Can authors also approach other human studies on the effects of dietary supplementation with selenium on women with gestational diabetes during pregnancy ? – if they couln’t be found, it should be appropriate to comment on this in the discussion section  If authors show this data, the conclusion of authors in this study will be more relevant.

3.     Please modify the font in the lines 395-398.

Thank you for your esteemed efforts in increasing our collective knowledge

Author Response

I have some comments on the article, which are listed below:

  1. This study presents the results of selenium supplementation in vitro conditions – this aspect should be mentioned in the title of the article

RESPONSE: The title has been amended as suggested. The new title is “Selenium prevents inflammation in human placenta and adipose tissue in vitro: Implications for metabolic diseases of pregnancy associated with inflammation”

  1. Can authors also approach other human studies on the effects of dietary supplementation with selenium on women with gestational diabetes during pregnancy ? – if they couldn’t be found, it should be appropriate to comment on this in the discussion section  If authors show this data, the conclusion of authors in this study will be more relevant.

RESPONSE: Please see the last paragraph were we have already mentioned the clinical studies. Lines 291-293

 “Our novel and exciting results are in agreement with clinical studies demonstrating low maternal selenium levels are associated with an increased risk of developing GDM [28, 29] and selenium supplementation in pregnant women with GDM improves glucose homeostasis, inflammation and oxidative stress [30].

  1. Please modify the font in the lines 395-398.

RESPONSE: Amended.

Thank you for your esteemed efforts in increasing our collective knowledge

Reviewer 2 Report

This Australian study investigates the effect of selenium supplementation on in vitro model of GDM and maternal obesity. The results indicate an anti-inflammatory effect of selenium and the authors postulate that selenium may have an important role in preventing inflammation associated with obesity and GDM. It is a well written manuscript with adequate methodology and clearly presented results. I have the following comments:

1.      What is the rate of obese and morbidly obese pregnant women in Australia? What is the rate of GDM among obese pregnant women? Please add these numbers in the introduction section.

2.      What is the role of human placental lactogen in the pathophysiology of GDM? Would selenium supplementation influence its effect on insulin resistance during pregnancy?

3.      The authors indicate that selenium supplementation in pregnant women with GDM improves glucose homeostasis (reference 30). What is the effect of selenium supplementation on neonatal outcomes in patients with GDM?

Author Response

  1. What is the rate of obese and morbidly obese pregnant women in Australia? What is the rate of GDM among obese pregnant women? Please add these numbers in the introduction section.

RESPONSE: The following sentence has been added to the first paragraph of the Introduction.In Australia, 15-20% of all pregnancies are complicated by GDM [4] while the incidence of GDM among obese pregnant women is up to 35% [5,6].’

[4] Australian Institute of Health and Welfare. "Incidence of gestational diabetes in Australia." (2019).

[5] George Mnatzaganian, Mark Woodward, H. David McIntyre, Liangkun Ma, Nicola Yuen, Fan He, Helen Nightingale, Tingting Xu & Rachel R. Huxley. Trends in percentages of gestational diabetes mellitus attributable to overweight, obesity, and morbid obesity in regional Victoria: an eight-year population-based panel study. BMC Pregnancy and Childbirth volume 22, Article number: 95 (2022)

[6] Cheney K, Farber R, Barratt AL, McGeechan K, de Vries B, Ogle R, et al. Population attributable fractions of perinatal outcomes for nulliparous women associated with overweight and obesity, 1990-2014. Med J Aust. 2018;208:119–25.

The reference list has been renumbered accordingly.

  1. What is the role of human placental lactogen in the pathophysiology of GDM? Would selenium supplementation influence its effect on insulin resistance during pregnancy?

      RESPONSE: Placental lactogen is thought to be involved in insulin resistance in normal pregnancy. Top our knowledge, there are no studies demonstrating a role for selenium supplementation on placental lactogen.

  1. The authors indicate that selenium supplementation in pregnant women with GDM improves glucose homeostasis (reference 30). What is the effect of selenium supplementation on neonatal outcomes in patients with GDM?

RESPONSE: The Reviewer raises an interesting point. However, we could find no studies that have reported on the effect of selenium supplementation on neonatal outcomes.